# Using electronic health records to quantify and stratify the severity of type 2 diabetes in primary care in England: rationale and cohort study design

Salwa S Zghebi,[1,2] Martin K Rutter,[3,4] Darren M Ashcroft,[5] Chris Salisbury,[6] Christian Mallen,[7] Carolyn A Chew-Graham,[7] David Reeves,[1] Harm van Marwijk,[8] Nadeem Qureshi,[9] Stephen Weng,[9] Niels Peek,[10] Claire Planner,[1] Magdalena Nowakowska,[1,2] Mamas Mamas,[11] Evangelos Kontopantelis[1,2]

For numbered affiliations see end of article.

**Correspondence to**
Dr Salwa S Zghebi;
salwa.zghebi@manchester.ac.uk

## ABSTRACT

**Introduction** The increasing prevalence of type 2 diabetes mellitus (T2DM) presents a significant burden on affected individuals and healthcare systems internationally. There is, however, no agreed validated measure to infer diabetes severity from electronic health records (EHRs). We aim to quantify T2DM severity and validate it using clinical adverse outcomes.

**Methods and analysis** Primary care data from the Clinical Practice Research Datalink, linked hospitalisation and mortality records between April 2007 and March 2017 for patients with T2DM in England will be used to develop a clinical algorithm to grade T2DM severity. The EHR-based algorithm will incorporate main risk factors (severity domains) for adverse outcomes to stratify T2DM cohorts by baseline and longitudinal severity scores. Provisionally, T2DM severity domains, identified through a systematic review and expert opinion, are: diabetes duration, glycated haemoglobin, microvascular complications, comorbidities and coprescribed treatments. Severity scores will be developed by two approaches: (1) calculating a count score of severity domains; (2) through hierarchical stratification of complications. Regression models estimates will be used to calculate domains weights. Survival analyses for the association between weighted severity scores and future outcomes—cardiovascular events, hospitalisation (diabetes-related, cardiovascular) and mortality (diabetes-related, cardiovascular, all-cause mortality)—will be performed as statistical validation. The proposed EHR-based approach will quantify the T2DM severity for primary care performance management and inform the methodology for measuring severity of other primary care-managed chronic conditions. We anticipate that the developed algorithm will be a practical tool for practitioners, aid clinical management decision-making, inform stratified medicine, support future clinical trials and contribute to more effective service planning and policy-making.

**Ethics and dissemination** The study protocol was approved by the Independent Scientific Advisory Committee. Some data were presented at the National Institute for Health Research School for Primary Care Research Showcase, September 2017, Oxford, UK and the Diabetes UK Professional Conference March 2018, London,

## Strengths and limitations of this study

► This is the first UK-based study to develop a diabetes severity scoring tool based on real-world electronic healthcare data to grade people with type 2 diabetes by their clinical severity.

► The study will include a large sample size and use high-quality medical data routinely collected from general practices with access to linked national hospitalisation and cause-specific mortality data sets.

► The association between the computed severity scores and future adverse outcomes (cardiovascular event, hospitalisation and mortality) will be used to validate the developed severity algorithm.

► There is a possibility to miss other severity indicators not recorded in the data sources used such as pharmacy dispensing data.

► Given that the linkage scheme is only available for consented general practices in England, the study cohort will be restricted to eligible patients registered within England (form nearly 58% of total Clinical Practice Research Datalink general practices).

UK. The study findings will be disseminated in relevant academic conferences and peer-reviewed journals.

## INTRODUCTION

The worldwide prevalence of diabetes mellitus is increasing with WHO estimating that over 422 million adults had diabetes in 2016 with a growth in the global prevalence from 4.7% in 1980 to 8.5% in 2014.[1 2] In the UK, the prevalence of type 2 diabetes mellitus (T2DM) has nearly doubled between 2004 and 2014.[3 4] People with diabetes have a higher risk for morbidity and mortality when compared with individuals without diabetes,[3 5] and hence the increasing prevalence presents a significant burden on healthcare resources. Diabetes management expenses are estimated to

BMJ

consume up to 11% of the total healthcare budget in the UK and USA.[6][7] Diabetes is a complex metabolic condition, of an increasing severity,[8][9] at individually varying levels, and progressive development of vascular complications and end organ damage over time. Diabetes is mainly managed in primary care in the UK,[10] overall Europe,[11] USA[12] and Asia.[13][14] In the UK, nearly 75% of the diabetes-associated costs relate to management of diabetes-related complications.[6]

The 'severity' of clinical conditions can be conceptualised as a progression of the underlying disease process. Increasing disease severity and development of associated complications lead to greater treatment complexity and clinical impact. The severity of chronic conditions, such as diabetes, has not been widely considered despite the clinical relevance and its likely impact on healthcare systems, where few studies have assessed disease severity among patients with T2DM or have quantified temporal trends or scores of severity.[15][16]

Severity scores could be clinically useful, particularly in T2DM; as such, a new summary score could contribute above what existing risk scores offer (eg, moving beyond risk for death) and also be highly relevant for use in six main clinical and research areas: (1) identifying complex patients with a higher need for future care; (2) identifying patients at early stages of disease (benchmarking); (3) providing information that will directly inform clinical care; (4) identifying trajectories in severity over time; (5) supporting research, for example, by serving as an important diabetes-specific covariate to consider in analyses, similarly to the Charlson Comorbidity Index, or as an outcome; and (6) providing data that will inform resource allocation in health systems, such as the National Health Service.

Here, we describe the conceptual development of a T2DM severity algorithm model, using electronic health records (EHRs) on clinical consultations and treatments. Our aim is to use routinely collected clinical and administrative data to develop a scoring tool which would quantify and grade the severity of T2DM. This model could potentially be used by clinicians to stratify patients based on disease severity and we aim to demonstrate potential advantages over similar risk scores. To validate the algorithm, our secondary aim is to examine the association of severity grades with risk of three main adverse outcomes: cardiovascular disease, hospitalisation and mortality. The developed severity stratification algorithm is anticipated to have direct impact on clinical practice and have wider implications on service planning and policy-making.

## METHODS AND ANALYSIS
### Data source
This study will use data from the Clinical Practice Research Datalink (CPRD). The CPRD is one of the world's largest EHR with anonymised primary care records from over 650 general practices over the UK.[17] Nearly 76% of all CPRD-registered patients are in England.[17]

CPRD data include clinical diagnoses, prescribed therapies, biochemical test results and referrals to healthcare services. The accuracy and completeness of diagnostic coding and validity of CPRD data, in clinical research, have been reported as excellent with positive predictive values (PPVs) of over 90% for nearly 14 conditions (including diabetes and cerebrovascular disease).[18] PPVs were defined as the proportion of CPRD diagnoses that were validated as true cases when compared with a gold standard such as general practitioner (GP) questionnaire or primary care records.

Two metrics for research quality data, recommended to be used by researchers, are available for CPRD records.[17] For patient-level data, the flag 'acceptable' indicates that a patient's record has met certain quality standards such as registration status, valid age and gender and record of patient events. For practice-level data, the up-to-standard (UTS) date is used as a data quality measure. The UTS date is calculated for each CPRD general practice as the latest date at which the general practice meets minimum quality criteria based on two central concepts: the continuity of recorded data and the number of recorded deaths, in comparison to an expected national range.[17][19]

Nearly 75% of general practices in England (approximately 58% of all CPRD practices) have consented to the CPRD Linkage Scheme for access to a number of linked data sets and national disease registries.[17] These include hospital data (including outpatient, admissions and accident and emergency (A&E) data), mortality records (held by the Office for National Statistics (ONS)), socioeconomic status and the cancer registry. The linked hospitalisation records, held by the Hospital Episode Statistics (HES), provide ethnicity data, admission and discharge dates, clinical diagnoses and procedures during hospitalisation recorded using the 10th revision of the International Classification of Diseases (ICD-10) and operating procedure codes. This study will only include patients registered with English general practices which provide UTS data and which participate in the CPRD Linkage Scheme.

### Study population
The study cohort will include individuals aged 35 years or over with T2DM (with ≥1 diagnostic code as in (online supplementary table S1) between 1 April 2007 and 31 March 2017. Patients with an ever record of type 1 diabetes (see online supplementary table S2) will be excluded unless they have records of non-insulin anti-diabetic therapies. We chose the study period after 2006 as the quality of CPRD data has improved substantially to adhere with the then introduced important changes to the national incentive scheme The Quality and Outcomes Framework (QOF), an incentivisation programme for all GP surgeries in the UK.[20][21] QOF exception reporting process allows general practices to exclude patients from indicators or a clinical domain based on discretionary exception codes. However, evidence on the use of exception codes has shown that they are being used

appropriately by practices and overall exception rates for diabetes patients are low.[22]

For this cohort construction, we will consider implementing previously validated algorithms designed to identify diabetes cases by avoiding potential misclassification in routinely collected data such as CPRD.[23 24] Eligible patients will be followed up until censored at the earliest instance of any of the following event dates: patient transferred out of the practice (any cause), last collection date for the practice, the study end on 31 March 2017 or death. The main demographic and clinical characteristics of the defined diabetes cohorts (such as age, gender, geographic region within England, patient-level and general practice-level social deprivation, body mass index (BMI) and baseline glycated haemoglobin ($HbA_{1c}$)) will be identified. Based on our previous studies, the expected sample size in a year will include 11 000 patients with T2DM registered with English general practices linked to the HES Admitted Patient Care data set, HES Outpatient and HES A&E data sets. A random 80% of the identified diabetes cohort (training dataset) will be used to develop the severity tool, with the remaining 20% of the cohort used as a validation data set, as described below.

### Severity domains

A systematic literature search for studies that developed algorithms or models to assess and quantify the severity of diabetes was conducted to identify the domains and subdomains for T2DM severity (to be published separately). Also, expert clinical opinion from members of the research team was used to supplement the search process and identify possible omissions. Clinical members in the team include pharmacists (DMA, SSZ), GPs (CS, CM, CACG, HVm and NQ), a consultant diabetologist (MKR) and a consultant cardiologist (MM), who used their expertise to create a list of relevant clinical domains for T2DM severity. The final domains to be included in the severity model will be decided during the analysis stage. Currently, the identified clinical domains that are relevant to the degree of progression of T2DM include: patient factors (diabetes duration (the period between T2DM diagnosis and the severity score estimation) and BMI), monitoring laboratory tests ($HbA_{1c}$ categories (threshold of 7% [53 mmol/mol]), and blood glucose levels), type of anti-diabetic therapy, other prescribed medications (such as lipid-regulating medications and ACE inhibitors (ACEIs)), comorbidities (including diabetes-related complications, depression), hospitalisation and surgical interventions. Comorbidities will be identified using appropriate code lists and contribute to the severity score according to each score computing method. The identified domains and subdomains are described in table 1. Domains will also be reviewed by a panel of 'experts by experience' (people with lived experience of T2DM) to provide patient validation of the severity scoring tool.

The clinical Read codes for the defined severity domains codes and the product codes for drug therapies will be identified using the (pcdsearch) Stata user command.[25]

The (pcdsearch) command is a search programme developed to extract code lists from typically very long lookup files associated with primary care databases using an input file containing a list of stubs for codes of interest to be searched for. For CPRD, the lookup files Medical data set is searched for all clinical Read codes and the Product lookup file, that includes unique product codes, is searched for all treatments. The corresponding ICD-10 codes will be used to identify clinical domains recorded in the hospitalisation data and ONS mortality records. The relevant CPRD operational identification entities for the laboratory tests will be identified. This is an ongoing process whereby the final lists of domains and codes will be reached by consensus among the clinical members of the team. All code lists will be available on the online clinical code repository (ClinicalCodes.org).[26]

### Study outcomes

The adverse outcomes of interest will be the development of the first event among cardiovascular disease (myocardial infarction, stroke), future hospitalisation (any hospitalisation, diabetes-related and cardiovascular hospitalisation) and death (diabetes-related mortality, cardiovascular mortality and all-cause mortality). Secondary outcome will be future hospitalisation due to hypoglycaemia, a relevant and potentially preventable adverse outcome. The hospitalisation and death outcomes will be identified using the linked HES and ONS mortality data, respectively. Similarly with severity domains, Read codes and ICD-10 codes (available on the ClinicalCodes.org online repository) will be used to identify the outcomes as appropriate.

### Statistical analyses

#### Diabetes severity algorithm

Using annual data bins and grouping diabetes patients in the training data set (include random 80% of the total diabetes cohort) from 1 April to 31 March between 2007 and 2017, the developed diabetes algorithm will grade the severity of T2DM using predefined (sub)domains. Our study period (after 2006) was selected to ensure very high data quality in primary care, while the addition of secondary care data will make our analyses even more robust, in terms of accurately classifying T2DM severity levels.

We will consider two approaches to derive numerical or categorical diabetes severity scores or levels. First, we will use a binary classification (severity indicator: present/absent) within each subdomain and calculate an aggregate score. The second approach, through hierarchical stratification of end organ microvascular and macrovascular complications, will involve increasing weighting within each subdomain, as severity increases where scores of 1, 2 or three on each subdomain will be assigned based on clinical input in terms of severity. Then, regression models, using death (primary outcome) and future hospitalisation (secondary outcome) as dependent variable, will be fitted from which the weights of its estimates will be used to calculate the weights for

**Table 1** The main (sub)domains identified to quantify the severity of type 2 diabetes

| | Severity domain | Severity subdomain |
|---|---|---|
| 1. | Risk factors* | ▶ Duration of type 2 diabetes[40]<br>▶ Body mass index (BMI)<br>▶ Hypertension<br>▶ Hyperlipidaemia<br>▶ Personal/Family history of cardiovascular disease<br>▶ Blood glucose levels<br> – Glycated haemoglobin (HbA$_{1c}$)[35]<br> – Fasting blood glucose (FBG) and random blood glucose (RBG) |
| 2. | Type/pattern of anti-diabetic treatment, insulin use and other therapies | ▶ Anti-diabetic therapy ever;[40] Therapies with cardiovascular benefits versus other; Changes in drug treatments;[43] or the number of prescribed treatments[42]<br>▶ Insulin use: prescription ever or within 1 year of diagnosis; Insulin initiation:[15] time to initiation<br>▶ Other therapies: ACE inhibitors (ACEI) and lipid-regulating therapies |
| 3. | Diabetes-related microvascular complications[15 34] | ▶ Neuropathy (foot ulcer, Charcot foot, gangrene, amputation)<br>▶ Nephropathy<br>▶ Retinopathy (laser therapy and blindness) |
| 4. | Renal disease | ▶ Microalbuminuria and proteinuria<br>▶ Moderate-severe chronic kidney disease (CKD) stages 3 and 4[34]<br>▶ End-stage renal disease (ESRD): kidney transplant and dialysis |
| 5. | Cardiovascular and cerebrovascular disease | ▶ Atherosclerosis[15 34]<br>▶ Myocardial infarction (MI)[15 34]<br>▶ Angina[15 34]<br>▶ Atrial/ventricular fibrillation (AF)/(VF)[34]<br>▶ Heart valve disease<br>▶ Heart failure (HF)[34]<br>▶ Peripheral vascular disease (PVD)[34]<br>▶ Transient ischaemic attack (TIA)<br>▶ Ischaemic stroke, haemorrhagic stroke[15 34] |
| 6. | Cardiovascular and cerebrovascular interventions | ▶ Coronary artery bypass graft (CABG)<br>▶ Coronary artery interventions (PCI/PTCA)<br>▶ Endovascular aneurysm repair (EVAR)<br>▶ PVD stenting and bypass procedures<br>▶ Heart valve interventions<br>▶ Use of defibrillator<br>▶ Carotid artery events, stenting and bypass interventions |
| 7. | Other comorbidities | ▶ Anxiety<br>▶ Depression<br>▶ Dementia<br>▶ Cognitive impairment |
| 8. | Hospital admissions | ▶ Any-cause hospital admissions<br>▶ Diabetes-attributable admission<br>▶ Cardiovascular disease-related admission |
| 9. | Emergency diabetes-related events | ▶ Hypoglycaemia<br>▶ Hyperosmolar hyperglycaemic state (HHS)[34]<br>▶ Diabetic ketoacidosis (DKA) or other coma[34] |

*Other demographic data (such as age, gender and the level of deprivation) are important predictors for adverse outcomes and will be included in the later risk prediction analysis.

severity subdomains. The highest possible severity score will be known when the final list of included domains and subdomains is decided. Hierarchical diagrams of the clinical severity of identified diabetes-related complications domains (figure 1), cerebrovascular domains (figure 2) and cardiovascular (figures 3 and 4,

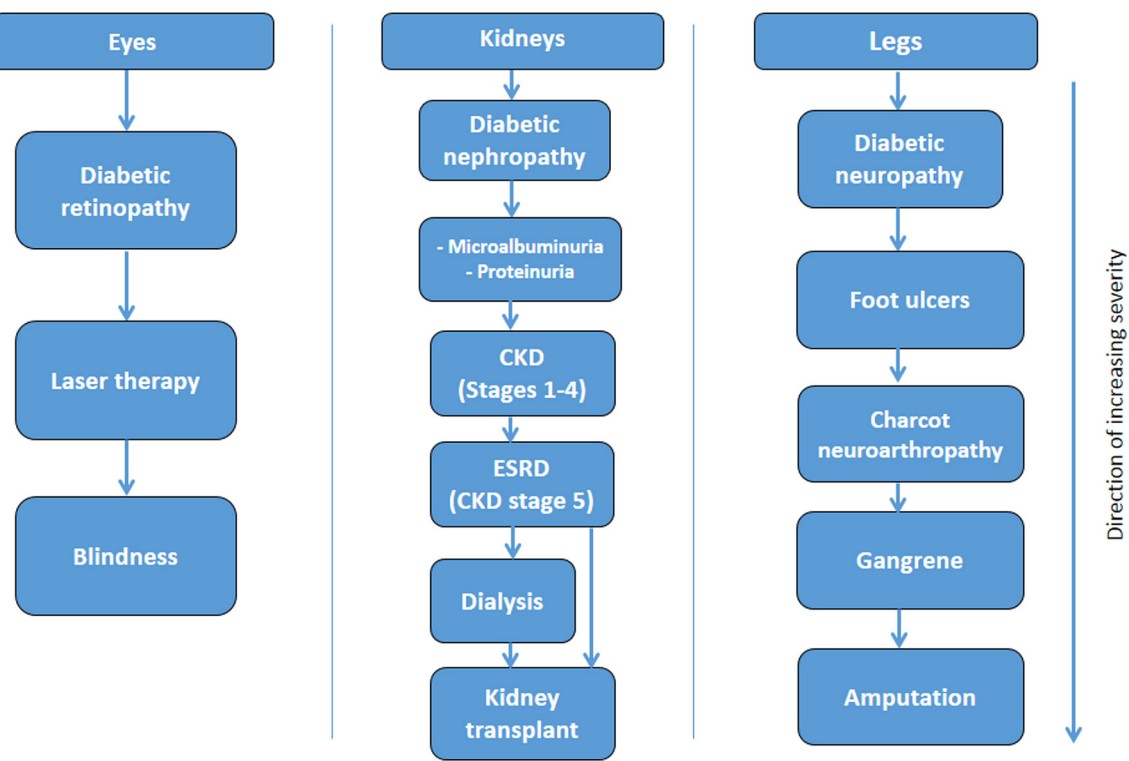

**Figure 1** Severity hierarchy of diabetes-related microvascular complications. CKD, chronic kidney disease; ESRD, end-stage renal disease.

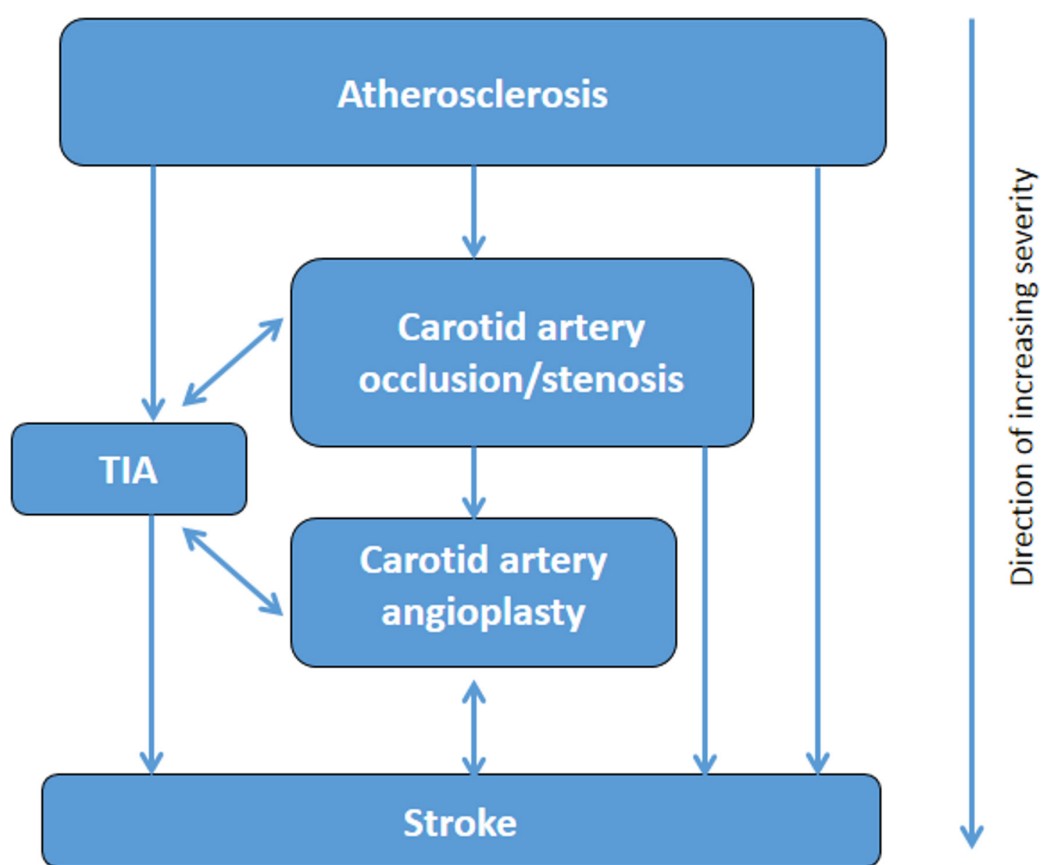

**Figure 2** Severity hierarchy of cerebrovascular domains. TIA, transient ischaemic attack.

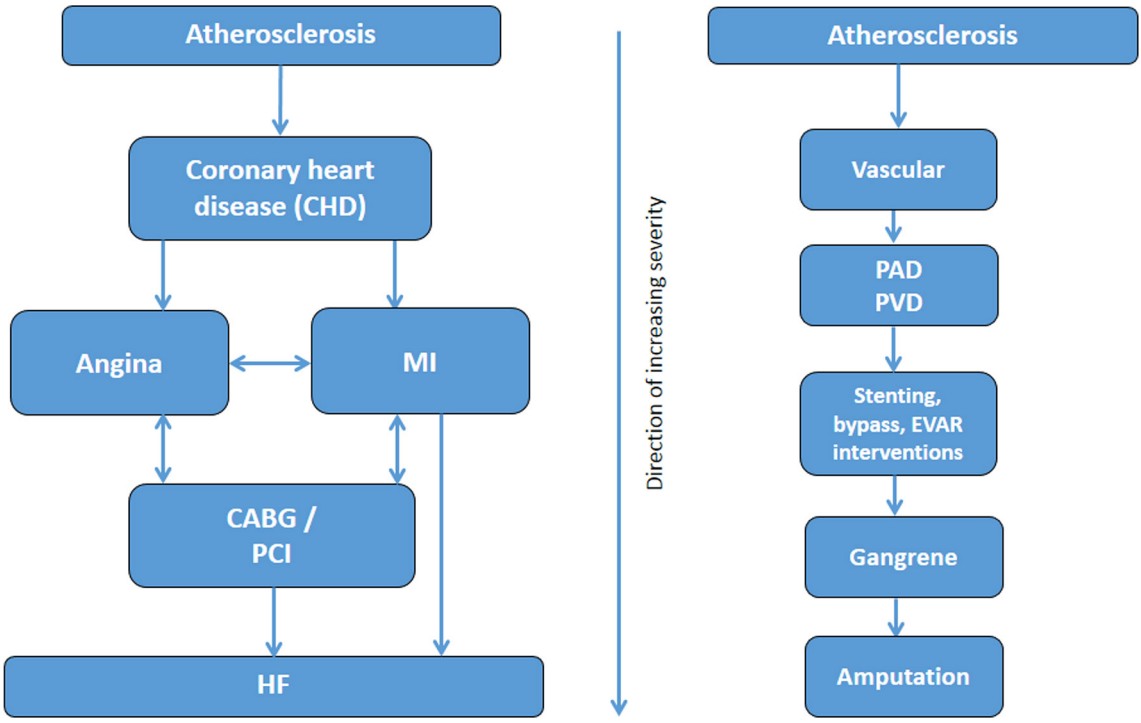

**Figure 3** Severity hierarchy of coronary and vascular domains. CABG, coronary artery bypass graft; EVAR, endovascular aneurysm repair; HF, heart failure; MI, myocardial infarction; PAD, peripheral arterial disease; PCI, percutaneous coronary intervention; PVD, peripheral vascular disease.

online supplementary figures S1 and S2) domains are presented.

Subdomains with many measurements over time, such as $HbA_{1c}$ levels and BMI, will be modelled as time-varying covariates (alternative models will be considered to account for non-linear changes). Within each time window, the average of these measurements will be used, and we will also use multiple imputation approaches to account for missing data. Provisionally, we will consider using data records within look-back periods between 3 and 5 years. Different look-back windows within this period will be tested to obtain the optimal time window. Descriptive statistics of the study population and the estimated scores will be performed.

### Severity algorithm validation

When we have developed a first version of the severity score tool, it will be validated statistically by quantifying the association of severity scores to future adverse outcomes. The primary outcome will be developing a cardiovascular event (myocardial infarction, stroke), future hospitalisation and death (diabetes-related mortality, cardiovascular

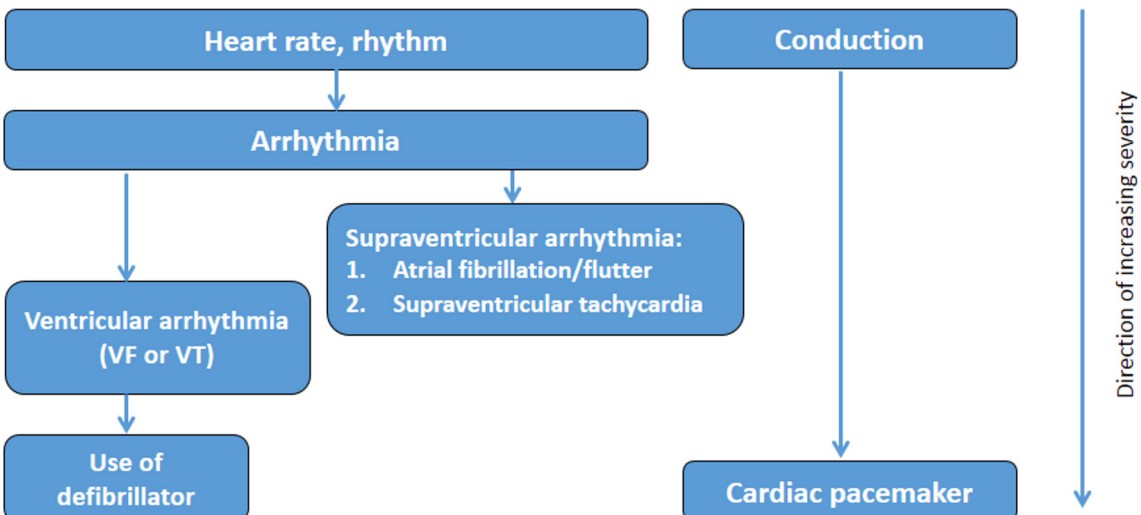

**Figure 4** Severity hierarchy of cardiovascular disease domains (heart rate, rhythm and conduction). VF, ventricular fibrillation; VT, ventricular tachycardia.

mortality and all-cause mortality). A secondary adverse outcome will be future hospital admission due to hypoglycaemia, a relevant and potentially preventable adverse outcome. Cox proportional hazards (after assessing proportional hazards assumption) and/or competing-risks regressions will be used to perform survival analyses and assess the relationship between the severity score and outcomes controlling for all available patient characteristics (such as age, gender, ethnicity and level of social deprivation). We aim to analyse and assess how severity scores differ across different levels of these variables. As stated earlier, a random 20% of the study cohort (validation data set) will be used to validate the performance of the severity algorithm that was developed in the training data set (80% of the diabetes cohort). Using the calculated subdomains weights (derived by the regression model estimates), Cox regression analysis will be used to validate the developed severity algorithm and we will assess the model performance using relevant measures (such as area under the receiver operating characteristic curve). Using this analysis, we will also test how much the 'sophisticated' full model that includes all relevant severity domains will add in terms of predictive validity of the adverse outcomes over a simpler model that includes demographic data (age, gender and level of social deprivation) alone.

### Longitudinal trends of diabetes severity

In patients newly diagnosed with T2DM, during the study period, we will assess the temporal trends of severity scores over time by calculating the time needed to progress along the quintiles of the severity scale starting from the date or year of diagnosis and identify the predictors to this progression across quintiles.

While the inclusion of some domains such as carotid artery stenting, the use of ACEIs or lipid-regulating drugs, kidney transplantation and laser therapy are clinically relevant and represent markers of greater level of disease severity, they also improve the prognosis (lessen the grade of severity), that is, raising a risk of 'feedback'. While we will further assess their role, we may have to not include these domains in the modelling at the analysis stage. All statistical analyses will be conducted using Stata Statistical Software for Windows: Release 15 (StataCorp LP, College Station, Texas, USA).

### Patient and public involvement and engagement

The public interest in the reuse of routinely collected electronic health data for research purposes has expanded over recent years. Our study presents a valuable opportunity to address the challenge of how to develop meaningful and productive patient and public involvement and engagement (PPIE) collaborations in observational studies that make use of secondary data.

We will identify suitable PPIE partners to collaborate on this study and develop creative strategies for meaningful involvement as the project develops. Participants to provide their input that will be particularly valuable for deciding what information would be most useful to patients with diabetes and for providing feedback into the developed diabetes algorithms.

In the latter stages of the study, jointly with our PPIE partners, we will plan and deliver a patients' and carers' workshop where we will present emerging findings and seek feedback to inform further work and the dissemination strategy. This will include an exercise around weights of domains within the algorithm. PPIE work will be reported using the GRIPP (Guidance for Reporting Involvement of Patients and Public) checklist.[27]

## DISCUSSION

The global and country-specific prevalence of diabetes has increased substantially.[2 28–31] For T2DM, prevalence rates have doubled over the past two decades.[4 32 33] Diabetes diagnosis is associated with higher risk for morbidity and mortality in comparison to people without diabetes.[5] Despite the natural progression of most chronic diseases, including T2DM, and their important clinical implication on prognosis and medical resources utilisation, only a few studies have attempted to grade the severity of chronic conditions over time. Our proposed study aims to develop a contemporary algorithm to quantify the severity of T2DM, a highly prevalent condition, using routine EHR. The scoring tool will be based on clinically relevant diagnoses and treatments modelled as severity domains and subdomains. The algorithm will enable clinicians to grade patients with T2DM according to the level of their disease severity, at baseline and longitudinally, as driven by the weights of the included clinical domains.

### Prior studies

Currently, there are limited data around validated severity measures that can be used in routine clinical practice managing patients with T2DM. Consequently, we are in parallel finalising a conducted systematic review for clinical-based diabetes severity models which will inform the refinement of this work. Here, we discuss three relevant diabetes severity measures that have been previously reported from countries outside the UK.

In the first study, Gini *et al* categorised the severity of T2DM (n=300) into four levels based on insulin use and the presence of diabetes-related complications (see online supplementary table S3).[15] A validation study in a random sample of cases was performed by interviewing their GPs. We, however, aim to include more patients with T2DM and consider more clinical data such as other coexisting conditions, non-insulin therapies, statins and ACEIs that have been shown to be associated with reduced risk of adverse outcomes.

In the second study, a diabetes symptom checklist was created to measure the perceived symptom severity and assess the changes over time in 185 patients with T2DM.[16] The checklist contained 34 items categorised by main clinical symptoms (see online supplementary table S3). The patterns of comorbidities and prescribed treatments

were reportedly associated with significant differences in the estimated severity scores.[16] The sample size was however relatively small and the checklist was based on patients' perception on diabetes symptoms severity.

Third, Young *et al* calculated a Diabetes Complications Severity Index (DCSI) in 4229 patients with type 1 or type 2 diabetes in one US geographic region to examine its association with adverse outcomes (risk of hospitalisation and mortality).[34] In comparison to using a simple numerical count of complications, DCSI found to be a better tool to predict adverse outcomes. The authors used pharmacy (insulin use only) and laboratory data to compute the severity index and included patients from clinics with largest ethnic diversity (see online supplementary table S3) . However, the DCSI missed additional domains such as diabetes duration, hyperlipidaemia and a wider range of diabetes-related and end organ damage manifestations that we aim to assess in addition to hospitalisation and mortality.

In the wider literature, although several studies have investigated the possible role of other various factors on the severity of diabetes, none have provided a severity scoring tool that uses data from various clinical domains as planned in our study that can use the wealth of information routinely collected in electronic healthcare databases. In these studies, approaches used to define diabetes severity included: comparing T2DM severity, before and after obesity surgery;[35] examining the association between diabetes severity and either haematological and immunological changes,[36] levels of urine citrate,[37] a biomarker for adverse outcomes;[38] grip strength;[39] or the use of complementary medicine to manage T2DM.[40]

Overall, the severity of T2DM has been previously assessed using the following domains: the complexity of anti-diabetic treatment regimens[35] or HbA$_{1c}$ levels;[37] a summary severity variable that includes vascular complications;[41] or a health status composite, number of comorbidities and patterns of treatments.[42 43] Other putative and less clinically robust indicators or animal models were reportedly used to assess the severity of T2DM such as the effect of different patient education approaches on diabetes severity;[44] evaluating the effect of parental history;[45] and the role of genetic,[46 47] metabolic[48] or inflammatory mediators.[49]

In comparison to our planned study, none of the previous studies included as many routinely collected clinically relevant variables as in our more inclusive summary severity score algorithm or assessed the association of estimated scores with the various adverse outcomes included in our defined primary and secondary endpoints.

### Potential strengths and limitations
Our proposed study has several potential strengths: First, it aims to present a contemporary measure to the available tools and the first UK-based study to develop an EHR-based severity scoring tool to grade patients by their T2DM severity. Second, the study uses high-quality

real-world medical data routinely collected from general practices. The use of routinely collected data indicates that severity scores can be generated automatically with minimal effort. Third, the views of PPIE collaborators will be incorporated in the development of the severity tool. Fourth, we will access two linked data sets: the hospitalisation (HES) data (to maximise data capture and reduce condition misclassification), and the cause-specific mortality data to ascertain causes of death. Fifth, the sample size is expected to be large enough to drive the development and evaluation of the severity algorithm. Sixth, a statistical validation of the developed algorithm is planned and described.

As we will use data available in CPRD and HES data sets, one of the limitations we anticipate is the possibility to miss other severity indicators not recorded in used data sets. These include detailed pharmacy data such as 'actual' dispensing and adherence data. Also, the use of routinely collected data is associated with missing values, being collected from questionnaires, and issues around the accuracy of coding. However, we plan to use appropriate imputation methods and definite criteria to minimise the effect of coding issues. A possible limitation that should be acknowledged is underestimated and poorly represented T2DM severity levels for patients not regularly attending a general practice, people missing appointments or patients not being reliably captured in the database due to very high mobility status (eg, homelessness). This limitation aligns with QOF exception reporting that allows practices to exclude patients from indicators or a clinical domain based on discretionary exception codes. However, the use of QOF exception coding was considered appropriate and its levels were very low, especially for informed dissent.[22] Another limitation is that our study will be restricted to patients registered with general practices in England, as CPRD currently only provides linkage to external national data sets for consented general practices in England. However, English general practices form the majority of all CPRD practices including nearly 76% of total registered patients.[17] Finally, due to project time constraints, it is not possible to validate the developed diabetes scores using questionnaires (as reported previously[15]) or replicating the algorithm in a separate data set. However, we have planned a statistical validation of the developed algorithm.

### Importance and the clinical implications of the severity tool
The developed algorithm and severity tool may have significant implications for primary care both in terms of disease management and resource allocation. Ideally, through future work of further validation and assessment of clinical utility, the severity tool will be of practical use in primary care through its implementation in the clinical computing systems used in the UK.[50] The clinical significance of the developed severity algorithm to primary care is driven by the inclusion of highly relevant clinical domains, such as diabetes-related complications and comorbidities, mapped to routinely collected data.

Additionally, by assessing the longitudinal patterns of severity, the developed tool may be more clinically relevant than the currently used proxy ($HbA_{1c}$), and thus it could be a more reliable indicator in informing practices' remuneration for diabetes care. Categorising individuals based on their diabetes severity will be relevant for risk stratification (which may enable safer delegation of care within the clinical team), help identify individualised patient risks and will help practitioners triage patients in need of a greater clinical input which informs towards stratified medicine to reduce future life-changing diabetes-related complications. Moreover, the weights of the severity scores may inform future clinical trials as the scoring tool considers a broader range of cardiovascular conditions than in most randomised clinical trials. Given the relatively low rates of cardiovascular outcomes in some trials, identifying patients with diabetes who are at higher risk via our severity algorithm would help to power trials with overall longer-term benefit for patients. Finally, the composite severity score may serve as an important confounding factor in future research, as an example, to match diabetes cases and controls in observational studies and clinical trials.

## ETHICS AND DISSEMINATION

Some data were presented at the annual National Institute for Health Research School for Primary Care Research (NIHR SPCR) Showcase, September 2017, Oxford, UK and at the Diabetes UK Professional Conference, March 2018, London, UK. The study findings will be disseminated in relevant academic conferences and peer-reviewed journals.

**Author affiliations**
[1]Division of Population Health, Health Services Research and Primary Care, School of Health Sciences, Faculty of Biology, Medicine and Health, Manchester Academic Health Science Centre (MAHSC), University of Manchester, Manchester, UK
[2]NIHR School for Primary Care Research, Centre for Primary Care, Manchester Academic Health Science Centre (MAHSC), University of Manchester, Manchester, UK
[3]Division of Diabetes, Endocrinology and Gastroenterology, School of Medical Sciences, Faculty of Biology, Medicine and Health, Manchester Academic Health Science Centre (MAHSC), University of Manchester, Manchester, UK
[4]Manchester Diabetes Centre, Manchester University NHS Foundation Trust, Manchester Academic Health Science Centre (MAHSC), Manchester, UK
[5]Division of Pharmacy and Optometry, School of Health Sciences, Faculty of Biology, Medicine and Health, Manchester Academic Health Science Centre (MAHSC), University of Manchester, Manchester, UK
[6]Centre for Academic Primary Care, Population Health Sciences, Bristol Medical School, University of Bristol, Bristol, UK
[7]Research Institute for Primary Care and Health Sciences, Keele University, Staffordshire, UK
[8]Division of Primary Care and Public Health, Brighton and Sussex Medical School, University of Brighton, Brighton, UK
[9]Division of Primary Care, School of Medicine, University of Nottingham, Nottingham, UK
[10]Division of Informatics, Imaging & Data Sciences (L5), School of Health Sciences, Faculty of Biology, Medicine and Health, Manchester Academic Health Science Centre (MAHSC), University of Manchester, Manchester, UK
[11]Keele Cardiovascular Research group, Centre for Prognosis Research, Institute for Primary Care and Health Sciences, Keele University, Stoke-on-Trent, UK

**Contributors** EK, SSZ, MM, MKR and HvM developed the study design and data analysis plan. SSZ, MM, MKR and HvM agreed on provisional clinical code lists. SSZ prepared the first draft of the manuscript, and EK, MM, MKR and HvM critically reviewed initial versions. CP contributed to the planned PPIE work. DR, CACG, CM, NP, DMA, CS, NQ, MN and SW reviewed and critically edited the manuscript. All authors approved the final version of the protocol before submission. SSZ is the guarantor.

**Funding** This study is funded by the National Institute for Health Research School for Primary Care Research (NIHR SPCR), grant number 331. This report is an independent research by the National Institute for Health Research.

**Disclaimer** The views expressed in this publication are those of the authors and not necessarily those of the NHS, the National Institute for Health Research or the Department of Health.

**Competing interests** EK, HvM, MM, DR, CACG, CP, CM, NP and MN declare no competing interests. SSZ reports support by the NIHR SPCR during this study. DMA has received grant funding from Abbvie and has served on advisory boards for Pfizer and GSK. MKR has received educational grant support from MSD and Novo Nordisk; has modest stock ownership in GSK; and has consulted for Roche. NQ reports grants from the NIHR SPCR during the conduct of the study. CS reports grants from NIHR SPCR during the conduct of the study; grants from NHS CLAHRC West, grants from Avon Primary Care Research Collaborative, outside the submitted work. SW serves as a member of the Clinical Practice Research Datalink Independent Scientific Committee (ISAC) at the UK Medicines and Health Regulatory Agency.

**Patient consent** Not required.

**Ethics approval** The CPRD's Independent Scientific Advisory Committee (ISAC) approved this study protocol.

**Provenance and peer review** Not commissioned; externally peer reviewed.

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
