## [Reviewer comments · BMJ Open]

ARTICLE DETAILS

TITLE (PROVISIONAL)	Using electronic health records to quantify and stratify the severity of type 2 diabetes in primary care in England: rationale and cohort study design
AUTHORS	Zghebi, Salwa; Rutter, Martin; Ashcroft, Darren; Salisbury, Chris; Mallen, Christian; Chew-Graham, Carolyn; Reeves, David; van Marwijk, Harm; Qureshi, Nadeem; Weng, Stephen; Peek, N; Planner, Claire; Nowakowska, Magdalena; Mamas, Mamas; Kontopantelis, Evangelos

VERSION 1 – REVIEW

REVIEWER	Rosa Gini Agenzia regionale di sanità della Toscana
REVIEW RETURNED	10-Jan-2018

GENERAL COMMENTS	The reserach question addressed by the protocol is relevant, however the study design suffers from some important weaknesses that should be addressed before initiating the study and clarified before publishing the protocol. The literature suggests that what GPs record in their medical records for the purpose of supporting their clinical activity and what they know about the patient is different: in [1] levels of severity for T2DM automatically detected from the medical records were not unfrequently misclassified, according to questionnaires of the GPs. Moreover, patients not seen by the GP may have their true severity level poorly represented in the medical records. In other words, even though PPV in CPRD in identifying conditions may be generally high, sensitivity may be imperfect, as it is strongly conditioned by the frequency of patient access to healthcare. This is particularly relevant in forming the dimension of monitoring laboratory tests. It is recommended to address those two challenges in the protocol. For instance, provisioning a specific analysis in the subpopulation of patents with apparetly low score and poor prognosis may be of help in identifying characteristics associated with poorly estimated severity score. From the Methods section (to my understanding), two types of processes are envisioned to generate the score: one ('first' and 'second' approach) is driven by clinical knowledge, the other is driven by predictivity. For the second type of score, the survival analysis does not provide a validation, because it is included in the development of the score (if I am understanding correctly). The third type of process, the decision tree, is unclear: what I understand is that a classification will be assumed (which?) and classification trees will be developed to create rules. Please clarify.
--

	Minor comments follow. ===== ABSTRACT From the abstract it is completely unclear how the score will be developed starting from the provisional and finalised set of domains. Please clarify shortly in the abstract. INTRODUCTION "providing research data such as serving as an important diabetes-specific covariate to consider in current research, similarly to the Charlson Comorbidity Index": the authors may consider that a variable capturing faithfully progression of diabetes may serve as an outcome, as well, or as an inclusion criterion, depending on the research question. LIMITATIONS th authors may want to mention that they have no possibility to validate their score using questionnaires or other forms of access to actual clinical condition of the patients. [1] Gini R, Schuemie MJ, Mazzaglia G, Lapi F, Francesconi P, Pasqua A, et al. Automatic identification of type 2 diabetes, hypertension, ischaemic heart disease, heart failure and their levels of severity from Italian General Practitioners' electronic medical records: a validation study. BMJ Open. 2016 Dec 1;6(12):e012413.
--	--

REVIEWER	Kim Rose Olsen Department of Public Health, University of Southern Denmark. Denmark
REVIEW RETURNED	11-Jan-2018

GENERAL COMMENTS	The authors present a protocol for the rationale and study design of an EHR based algorithm to quantify and stratify the severity of type 2 diabetes in primary care. The objective of the study is valid and there is not a lot of literature on the subject. Hence the study will definitely contribute to the literature. However I have several main concerns with the paper being submitted as a protocol:  - Is it in general terms necessary to publish protocols on this type of work? Developing an algorithm for a severity score is basically about finding a model that, in the best possible way, explain occurrence of the identified outcomes for validating the model. Hence there is a clear guidance to inform the study design – namely to choose the model that have the highest explanatory power of outcome events. Publishing a protocol may have two purposes; to avoid that the researches strategically adapt study designs that give "favourable" results and to inform other researchers about ongoing work to avoid duplication. I have problems judging whether the first purpose it fulfilled with regards to the current paper. - The information of important elements in the protocol is not detailed enough to make it out for a protocol. Examples:  o Outcomes for validation are a composite endpoint of Cardiovascular events, hospitalizations and death. Pls list ICD10
---

	codes to define Cardiovascular events and hospitalization (both inpatient and outpatient??).  o Severity domains include diabetes duration, Hba1c, microvascular and comorbidities. How are diabetes duration measured? What is the threshold for Hba1c used to inform severity? How are microvascular complications defined? Which comorbidities are included and how are they defined? o Ambulatory Care sensitive conditions is chosen as secondary outcomes, but no definitions is supplied – neither as to which ICD10 codes are used for individual therapeutic areas (diabetes, CVD, etc) in the definition or as to how many therapeutic areas are included (is it only diabetes and CVD or also COPD etc?). This is problematic as quite a lot of definitions exists. o P.6 In 47 ” The main demographic and clinical characteristics of the defined diabetes cohorts will be identified” Such variables should be defined in a protocol o P.7 In 36/37 ” The clinical Read codes for the defined severity domains codes and the product codes for drug therapies will be identified using the [pcdsearch] Stata command”: Pls elaborate. I don’t know this command and readers in the same situation have no chance assessing whether this is appropriate. - Data are available for the period 2006-2016. The study population is defined in 2006 and followed until censored or end of study period. An algorithm on severity may gain a lot from knowing a lot about the history of the patients – e.g. occurrence of the chosen outcome events (CVD and hospitalizations) before the index date. Why not identify the population in 2010/11 and the both look back and forward??
--	---

VERSION 1 – AUTHOR RESPONSE

Reviewer: 1

Reviewer Name: Rosa Gini

Institution and Country: Agenzia regionale di sanità della Toscana

The reserach question addressed by the protocol is relevant, however the study design suffers from some important weaknesses that should be addressed before initiating the study and clarified before publishing the protocol.

1. The literature suggests that what GPs record in their medical records for the purpose of supporting their clinical activity and what they know about the patient is different: in [1] levels of severity for T2DM automatically detected from the medical records were not unfrequently misclassified, according to questionnaires of the GPs. Moreover, patients not seen by the GP may have their true severity level poorly represented in the medical records. In other words, even though PPV in CPRD in identifying conditions may be generally high, sensitivity may be imperfect, as it is strongly conditioned by the frequency of patient access to healthcare. This is particularly relevant in forming the dimension of monitoring laboratory tests. It is recommended to address those two challenges in the protocol. For instance, provisioning a specific analysis in the subpopulation of patents with apparetly low score and poor prognosis may be of help in identifying characteristics associated with poorly estimated severity score.

[1] Gini R, Schuemie MJ, Mazzaglia G, Lapi F, Francesconi P, Pasqua A, et al. Automatic identification of type 2 diabetes, hypertension, ischaemic heart disease, heart failure and their levels of severity from Italian General Practitioners’ electronic medical records: a validation study. *BMJ Open*. 2016 Dec 1;6(12):e012413.

Response:

Thank you for this important comment.

First challenge - EMR misclassified T2DM severity levels vs. GP Questionnaires as in Gini et al. 2016:

- We chose the study period from 2006 as the quality of CPRD data has improved substantially to adhere with the then introduced important changes to the national incentive scheme The Quality and Outcomes Framework (QOF). The QOF, is an annual reward and incentive programme for all GP surgeries in England detailing practice achievement results. Although voluntary, the QOF saw complete participation from general practices, due to the financial incentives provided which supplemented up to 25% of GP salaries. These revisions included specific changes to diabetes by adding new indicators (set of achievement measures) for diabetes, and also helped standardise the recording of diabetes cases in clinical computer systems (e.g. clarifying between type 1 and type 2 forms).
- Also, as listed in Table 1, our algorithm will include more clinical variables than insulin use and the listed diabetes-related complications (as used in the important study that we have already cited for its high relevance(1)). In addition, we will be using secondary records to supplement our clinical data capture. Therefore, we anticipate that the planned approach would minimise misclassification of the calculated severity levels.
- The diabetes severity misclassification reported by Gini et al. 2016:
 - The misclassification was found in levels 1 and 3 that is involving absence/presence of diabetes complications. Our access to hospitalisation data will improve the capture of these complications.
 - As stated by the authors, some of their results are likely specifically related to the format of the used primary care medical record database HSD.

Second challenge - records conditioned by the frequency of patient access to the healthcare facility:

- We are using real-world medical data, as in many observational studies, which is affected by patients' frequency of visits to their GPs. As outlined in our edited "Limitations" section, the possibility of missing severity indicators because are not recorded is acknowledged. In addition, missing these indicators due to patients not attending a general practice, is indeed a possibility (and more likely in our view to non-recording by GPs following attendance, since UK primary care is completely computerised and has been since before the introduction of the QOF). However, it is also a fact that approximately 99% of the UK population is registered with a general practice, since the NHS is a free healthcare system at the point of access. Nevertheless, we will be missing records on people who are very mobile (e.g. are homeless), a fact that is also acknowledged in the paper. Specifically for diabetes, however, due to the nature of the condition, we feel it is highly unlikely for patients with diagnosed diabetes (our cohort) not to visit primary care (especially since general practices are incentivised to see these patients at least once every year), with the potential exception of special groups as previously discussed.
- Regarding planning a specific analysis, thank you this is an excellent suggestion. A separate analysis of the patients with apparent low and high severity scores will be considered to help identify the characteristics of patients at the two boundaries of the diabetes severity scale.

Changes:

- Lines 39-45 now added to the "Study Population" section (Page 6) to account for the first comment:

"We chose the study period from 2006 as the quality of CPRD data has improved substantially to adhere with the then introduced important changes to the national incentive scheme The Quality and Outcomes Framework (QOF), an incentivisation programme for all GP surgeries in the UK.(2,3)"

- Lines 9-14 added to the "Diabetes Severity Algorithm" section (Page 9):

"Our study period (2006 onwards) was selected to ensure very high data quality in primary care, while the addition of secondary care data will make our analyses even more robust, in terms of accurately classifying T2DM severity levels."

- Line 44 (Page 14) in the “Limitations” section in the manuscript now includes the following text as a response to the reviewer’s comment:

“In addition, underestimating T2DM severity levels for patients not regularly attending a general practice or not being reliably captured in the database due to very high mobility (e.g. homeless patients), is a possibility and should be acknowledged”

1. Gini R, Schuemie MJ, Mazzaglia G, Lapi F, Francesconi P, Pasqua A, et al. Automatic identification of type 2 diabetes, hypertension, ischaemic heart disease, heart failure and their levels of severity from Italian General Practitioners’ electronic medical records: a validation study. *BMJ open*. 2016;6(12):e012413. Epub 2016/12/13.
2. Lester H, Campbell S. Developing Quality and Outcomes Framework (QOF) indicators and the concept of ‘QOFability’. *Quality in Primary Care*. 2010;18:103-9.
3. Calvert M, Shankar A, McManus RJ, Lester H, Freemantle N. Effect of the quality and outcomes framework on diabetes care in the United Kingdom: retrospective cohort study. *BMJ*. 2009;338:b1870.

2. From the Methods section (to my understanding), two types of processes are envisioned to generate the score: one (‘first’ and ‘second’ approach) is driven by clinical knowledge, the other is driven by predictivity. For the second type of score, the survival analysis does not provide a validation, because it is included in the development of the score (if I am understanding correctly). The third type of process, the decision tree, is unclear: what I understand is that a classification will be assumed (which?) and classification trees will be developed to create rules. Please clarify.

Response:

The survival analysis will be based on outcomes developed after the calculation of the score i.e. not using any records used in the development of the severity score. The same would apply for the second type of score (hierarchical), just that the numerical value of the score will increase as the subdomain increases in severity.

In response to this important comment, we made it clearer in the manuscript to explain our plan to use a validation set of training (to develop the scoring tool) and validation datasets. A random 10% subset will be used as a validation dataset.

Regarding the third type of process, the decision tree, the protocol has now been amended to only include the first and second approaches to calculate severity scores and to drop the decision tree method.

Changes:

- The decision tree approach (page 9) is now dropped from the “Methods” section in the manuscript.
- New Lines 35-38 (Page 10) added to the “Severity Algorithm Validation” section: “The cohort will be randomly split into a training (90%) and validation dataset (10%), and the latter will be used to validate the performance of the severity algorithm.”

Minor comments follow.

3. ABSTRACT

From the abstract it is completely unclear how the score will be developed starting from the provisional and finalised set of domains. Please clarify shortly in the abstract.

Response:

Thank you for this comment. The "Methods and Analysis" section in the Abstract now describes how the severity score will be developed. However, this had to be edited strictly given that the original abstract had the maximum allowed word limit of 300 words.

Change:

The added text to the Abstract (Lines 24-26) reads as follows:

"Severity scores will be developed by two approaches: i) calculating an aggregate score of severity domains; ii) through hierarchical stratification of complications."

4. INTRODUCTION

"providing research data such as serving as an important diabetes-specific covariate to consider in current research, similarly to the Charlson Comorbidity Index": the authors may consider that a variable capturing faithfully progression of diabetes may serve as an outcome, as well, or as an inclusion criterion, depending on the research question.

Response:

Thank you for this important suggestion.

Change:

The relevant text in the Introduction (Page 4) has now been changed and reads as follow:

"v) supporting research, for example, by serving as an important diabetes-specific covariate to consider in analyses, similarly to the Charlson Comorbidity Index, or as an outcome."

5. LIMITATIONS

th authors may want to mention that they have no possibility to validate their score using questionnaires or other forms of access to actual clinical condition of the patients.

Response:

Thank you for this comment.

Change:

In response, the following text was added to the "Limitations" section (Page 15):

"Finally, due to project time constrains, there is no possibility to validate the developed diabetes scores using questionnaires (as reported previously (1)) or replicating the algorithm in a separate dataset. However, we have planned a statistical validation of the developed algorithm".

1. Gini R, Schuemie MJ, Mazzaglia G, Lapi F, Francesconi P, Pasqua A, et al. Automatic identification of type 2 diabetes, hypertension, ischaemic heart disease, heart failure and their levels of severity from Italian General Practitioners' electronic medical records: a validation study. *BMJ open*. 2016;6(12):e012413. Epub 2016/12/13.

Reviewer: 2

Reviewer Name: Kim Rose Olsen

Institution and Country: Department of Public Health, University of Southern Denmark. Denmark

The authors present a protocol for the rationale and study design of an EHR based algorithm to quantify and stratify the severity of type 2 diabetes in primary care. The objective of the study is valid and there is not a lot of literature on the subject. Hence the study will definitely contribute to the literature. However I have several main concerns with the paper being submitted as a protocol:

1. Is it in general terms necessary to publish protocols on this type of work? Developing an algorithm for a severity score is basically about finding a model that, in the best possible way, explain occurrence of the identified outcomes for validating the model. Hence there is a clear guidance to inform the study design – namely to choose the model that have the highest explanatory power of outcome events. Publishing a protocol may have two purposes; to avoid that the researches strategically adapt study designs that give "favourable" results and to inform other researchers about ongoing work to avoid duplication. I have problems judging whether the first purpose it fulfilled with regards to the current paper.

Response:

Thank you for your positive comments on the potential contribution of our study to the literature. Firstly, we would like to clarify that this paper mainly serves as a vehicle for us to agree on the approach, since this is a complex piece of work. And on the basis of that, we thought it was worthy pursuing publication to provide the process through which such tools can be developed to other researchers. So this is not really an RCT type protocol, with a primary purpose of preventing the cherry picking of results. The second reason is our aim to use this work to facilitate the future publications that will stem from this programme of work, with interested readers being able to access this paper for complete clarity on the methods. However, we will adhere to what we have stated in the plan. So the reviewer is right in having difficulty placing this work in a set research space, and we hope we have clarified this now. To better reflect on this, we have edited the title of the manuscript to "Using electronic health records to quantify and stratify the severity of type 2 diabetes in primary care in England: rationale and cohort study design".

2. The information of important elements in the protocol is not detailed enough to make it out for a protocol. Examples:

2.1. Outcomes for validation are a composite endpoint of Cardiovascular events, hospitalizations and death. Pls list ICD10 codes to define Cardiovascular events and hospitalization (both inpatient and outpatient??).

Response:

Thank you for this comment. The code lists are considerably long to be tabulated in the main manuscript. Therefore, and as we have stated in the original manuscript, the Read and ICD-10 code lists of the cardiovascular outcomes and hospitalisation codes will be available on the online clinical code repository (ClinicalCodes.org) that was developed by some members of our team.

The linked hospitalisation events linked to CPRD data are recorded using ICD-10 coding, while Read codes are used for CPRD data events recorded in general practices. At this stage, we have access to the linked inpatient dataset approved (access to outpatient records still pending).

Change:

In response to this comment, the text in Line 41 (Page 8) was edited for clarity and reads as: "Similarly with severity domains, Read codes and ICD-10 codes (available on ClinicalCodes.org online repository) will be used to identify the outcomes as appropriate."

2.2. Severity domains include diabetes duration, Hba1c, microvascular and comorbidities. How are diabetes duration measured? What is the threshold for Hba1c used to inform severity? How are microvascular complications defined? Which comorbidities are included and how are they defined?

Response:

Thank you for raising these relevant questions.

- Diabetes duration will be calculated as the period between T2DM diagnosis and the severity score estimation.
- HbA1c Categories will be used with a threshold of 7% (53 mmol/mol).
- Microvascular complications will be identified from primary care and secondary care datasets using relevant clinical code lists (to be uploaded on ClinicalCodes.org). Overall, events to be defined as binary (present/absent) and contribute to the score according to each score computing method as described in details in the “Methods” section.
- Comorbidities will include diabetes-related microvascular complications, cardiovascular and cerebrovascular disease (including surgical interventions), renal disease (from microalbuminuria to ESRD), mental health conditions (such as anxiety, depression, dementia and cognitive impairment). A detailed list of all comorbidities of interest is described in Table 1 (Page 21). As planned for microvascular complications, recorded comorbidities will be identified using appropriate code lists and will be defined according to each method as described in details in the “Methods” section.

Changes:

Relevant sections of the protocol have been edited to address the reviewer’s comment as follows:

- Lines 35-36 (Page 7): diabetes duration (the period between T2DM diagnosis and the severity score estimation)
- Lines 38-39 (Page 7): monitoring laboratory tests (glycated haemoglobin (HbA1c) categories (threshold of 7% [53 mmol/mol])).
- Lines 44-45 (Page 7): Comorbidities will be identified using appropriate code lists and contribute to the severity score according to each score computing method.

2.3. Ambulatory Care sensitive conditions is chosen as secondary outcomes, but no definitions is supplied – neither as to which ICD10 codes are used for individual therapeutic areas (diabetes, CVD, etc) in the definition or as to how many therapeutic areas are included (is it only diabetes and CVD or also COPD etc?). This is problematic as quite a lot of definitions exists.

Response:

Thank you. Our defined secondary outcome is future hospitalisation due to hypoglycaemia and this is stated in the paper (Page 8 and Page 9).

2.4. P.6 In 47 ” The main demographic and clinical characteristics of the defined diabetes cohorts will be identified” Such variables should be defined in a protocol

Response:

These variables to include age, gender, geographic region within England, patient-level and general practice-level social deprivation, BMI, and baseline HbA1c.

Change:

Lines 9-11 on Page 7 have been added in a response to this comment and read as follows:

"The main demographic and clinical characteristics of the defined diabetes cohorts (such as age, gender, geographic region within England, patient-level and general practice-level social deprivation, BMI, and baseline HbA1c) will be identified."

2.5. P.7 In 36/37 " The clinical Read codes for the defined severity domains codes and the product codes for drug therapies will be identified using the [pcdsearch] Stata command": Pls elaborate. I don't know this command and readers in the same situation have no chance assessing whether this is appropriate.

Response:

Thank you, we had originally provided a reference for interested readers. However, we do agree with the reviewer that a brief description of the used search Stata command would be helpful. Importantly, the resulted code lists are reviewed by clinical experts within the team afterwards to check the appropriateness and relevance of the retrieved codes.

The statistical program was published in a relevant Plos One paper: Olier I, Springate DA, Ashcroft DM, Doran T, Reeves D, Planner C, et al. Modelling Conditions and Health Care Processes in Electronic Health Records: An Application to Severe Mental Illness with the Clinical Practice Research Datalink. Plos One. 2016;11(2).

Change:

The following text has been added to the "Severity Domains" section Lines 8-16 (Page 8):

"The [pcdsearch] command is a search programme developed to extract code lists from typically very long lookup files associated with primary care databases using an input file containing a list of stubs for codes of interest to be searched for. For CPRD, the lookup files Medical dataset is searched for all clinical Read codes and the Product lookup, file that includes unique product codes, is searched for all treatments."

3. Data are available for the period 2006-2016. The study population is defined in 2006 and followed until censored or end of study period. An algorithm on severity may gain a lot from knowing a lot about the history of the patients – e.g. occurrence of the chosen outcome events (CVD and hospitalizations) before the index date. Why not identify the population in 2010/11 and the both look back and forward??

Response:

Thank you for this worthy comment. As mentioned in the manuscript, we aim to estimate the diabetes severity at baseline, this of course will be developed using pre-index clinical records. Different lengths of look-back windows will also be examined. This is very relevant given the nature of T2DM which frequently remains undiagnosed for many years whereas early hyperglycaemia-induced clinical changes (related to complications) begin to develop before clinical diagnosis of T2DM.

Regarding the suggestion to identify a diabetes population in 2010/11, we aim to look at a wider interval as possible rather than a single cohort. Importantly, we chose the start year 2006 as the quality of CPRD data has improved substantially from 2006 onwards after the introduction of important changes to the national incentive scheme Quality and Outcomes Framework (QOF) (a scheme that rewarded general practices in England for providing high-quality care based on defined clinical indicators).

However, we may conduct the suggested approach on a single mid-point cohort for patient population in 2010/2011 and look at nearly equal pre- index and post-index time windows.

FORMATTING AMENDMENTS (if any)

Required amendments will be listed here; please include these changes in your revised version:

VERSION 2 – REVIEW

REVIEWER	Rosa Gini Agenzia regionale di sanità della Toscana
REVIEW RETURNED	14-Feb-2018

GENERAL COMMENTS	I thank the authors for addressing my main comments in their revisions. However, I think some more clarifications would improve the protocol and the consequent study. Since this is the objective of publishing the protocol in the first place, I hope the authors meet this chance to further improve it. I still don't see why the fact that a patient diagnosed with T2DM is registered with a GP would force him/her to attend the general practice for an insidious disease such as T2DM, which does not give symptoms for long periods. Unfortunately some patients, even though they have a regular home, prefer to forget their condition and discontinue their medications and/or diagnostic follow-up, no matter how persuasive their GP is. Indeed, in QOF exception codes are foreseen for patients who refuse appointments, see (Campbell 2011). I suggest the authors to address this case in their analysis, or to explicitly mention this as a limitation. (Campbell 2011) Campbell S, Hannon K, Lester H. Exception reporting in the Quality and Outcomes Framework: views of practice staff – a qualitative study. Br J Gen Pract. 2011 Apr 1;61(585):e183–9. I thank the authors for clarifying in the rebuttal letter that the survival model will be run on a random 90% of the cohort and used to develop the score, which will be then validated for prediction on the remaining 10%. However from the protocol it's still unclear to me whether the survival model will be used to weight the score, or not. In the abstract the use of this model to develop the score is not mentioned. In the methods section, however, in the subsection 'Diabetes Severity Algorithm' there is a mention of a 'model' that will be used to weight sub-domains, but I could find no mention of what this model is in the previous parts of the protocol, nor of the fact that it will be run on a sample or any other detail. In the next subsection 'Severity Algorithm Validation' the models are finally described, but they are introduced as a means for validation, not for development. At some point the splitting of the cohort in two samples is introduced, but it is unclear why and how the 90% of the cohort will be used. I suggest to rephrase completely the two subsections and describe the activities that will be performed in the chronological and logical order.
--

VERSION 2 – AUTHOR RESPONSE

Reviewer: 1

Reviewer Name: Rosa Gini

Institution and Country: Agenzia regionale di sanità della Toscana

1. I thank the authors for addressing my main comments in their revisions. However, I think some more clarifications would improve the protocol and the consequent study. Since this is the objective of publishing the protocol in the first place, I hope the authors meet this chance to further improve it.

I still don't see why fact that a patient diagnosed with T2DM is registered with a GP would force him/her to attend the general practice for an insidious disease such that T2DM, which does not give symptoms for long periods. Unfortunately some patients, even though they have a regular home, prefer to forget their condition and discontinue their medications and/or diagnostic follow-up, no matter how persuasive their GP is. Indeed, in QOF exception codes are foreseen for patients who refuse appointments, see (Campbell 2011). I suggest the authors to address this case in their analysis, or to explicitly mention this as a limitation.

(Campbell 2011) Campbell S, Hannon K, Lester H. Exception reporting in the Quality and Outcomes Framework: views of practice staff – a qualitative study. *Br J Gen Pract.* 2011 Apr 1;61(585):e183–9.

Authors' response:

Firstly, we would like to thank the reviewer for the time reviewing our paper for the second time and for providing this useful feedback to help improve our protocol and planned analysis.

As outlined in our methodology, we will be using medical data that is routinely-collected in primary care. As in many observational studies, using real-world clinical data is inevitably affected by patients' frequency of visits to their GPs. However, it is important to note that our analyses will also use secondary care data which will increase the capture of related events in this patient population. In addition, we had already acknowledged those limitations (Page 14) in two points, indicated by firstly stating a possibility to miss other severity indicators not recorded in used dataset such as dispensing and adherence data (this relates to what the reviewer has pointed out as some T2DM patients who "discontinue their medications"); secondly, that a possible underestimation of T2DM severity levels for patients not regularly attending a general practice is also acknowledged. But, in response to the reviewer comments, we have edited this protocol section to highlight these limitations.

We also thank the reviewer for the suggested reference by Campbell et al. (2011) on QOF exception reporting. We are aware of the role of exception reporting and the senior author has worked on such aspect of the QOF in the past, please see <http://www.bmj.com/content/344/bmj.e2405> and <http://qualitysafety.bmj.com/content/25/9/657>. We are now discussing the role of exception reporting in the protocol. Evidence on the use of exception codes has shown that they are being used appropriately by practices and overall exception rates for diabetes patients are low, while informed dissent exception rates (the rates the reviewers primarily discuss) are very low (see the first paper, Table 1). For example, the highest informed dissent exception rate was 2.5% for HbA1c control,

which is arguably the most invasive of the QOF indicators – rates were much lower for other indicators. We are now including this paper in the 'Discussion' and continue to argue that some non-compliant patients will be missed and the severity score for them will be underestimated, but these people are not many.

Changes:

In response to the reviewer's comment, the following statements have been added to the 'Methods' and the 'Limitations' section for more clarity and they read as:

i. The 'Methods' subsection

"QOF exception reporting process allows general practices to exclude patients from indicators or a clinical domain based on discretionary exception codes. However, evidence on the use of exception codes has shown that they are being used appropriately by practices and overall exception rates for diabetes patients are low.(1)"

ii. The 'Limitations' subsection

"A possible limitation that should be acknowledged is underestimated and poorly represented T2DM severity levels for patients not regularly attending a general practice, people missing appointments or not being reliably captured in the database due to very high mobility status (e.g. homelessness). This limitation aligns with QOF exception reporting that allows practices to exclude patients from indicators or a clinical domain based on discretionary exception codes. However, the use of QOF exception coding was considered appropriate and its levels were very low, especially for informed dissent.(1)"

1. Doran T, Kontopantelis E, Fullwood C, Lester H, Valderas JM, Campbell S. Exempting dissenting patients from pay for performance schemes: retrospective analysis of exception reporting in the UK Quality and Outcomes Framework. *BMJ*. 2012;344:e2405.

2. I thank the authors for clarifying in the rebuttal letter that the survival model will be run on a random 90% of the cohort and used to develop the score, which will be then validated for prediction on the remaining 10%. However from the protocol it's still unclear to me whether the survival model will be used to weight the score, or not. In the abstract the use of this model to develop the score is not mentioned. In the methods section, however, in the subsection 'Diabetes Severity Algorithm' there is a mention of a 'model' that will be used to weight sub-domains, but I could find no mention of what this model is in the previous parts of the protocol, nor of the fact that it will be run on a sample or any other detail. In the next subsection 'Severity Algorithm Validation' the models are finally described, but they are introduced as a means for validation, not for development. At some point the splitting of the cohort in two samples is introduced, but it is unclear why and how the 90% of the cohort will be used. I suggest to rephrase completely the two subsection and describe the activities that will be performed in the chronological and logical order.

Authors' response:

Thank you for this feedback. We have made changes to the protocol in response to the points raised in the comment above. Below, is a detailed point-by-point response ordered as the points were listed in the comment, followed by the changes made to the paper:

- Yes, in the survival models we plan to use severity domains weights (as calculated from the regression model) to assess the association between severity weights (severity score) and adverse outcomes. Apologies if that was not satisfactorily clear in the protocol. This use of survival models is now stated more explicitly in both the 'Abstract' (subject to word limit) and in the 'Severity Algorithm Validation' sections in the paper.
- Regression model to be used to develop the severity score (as domains weight) is now mentioned in the 'Abstract' (subject to word limit).
- The regression model is firstly described in the 'Diabetes Severity Algorithm' as we considered that is likely more appropriate to be described in this subsection compared to earlier subsections of the 'Methods'. But, the reviewer is right about this observation and the model used to develop the severity tool is now stated earlier in the protocol 'Abstract'. Also, the plan to randomly split the study cohort so to develop the severity scores in 90% of the sample is now introduced earlier in both the 'Diabetes Severity Algorithm' and 'Study Population' subsections for more clarity as suggested by the reviewer.
- After the revisions made in response to this review, our plan for random split of the identified diabetes cohort into two sections is now described in earlier sections in the protocol and not appears in the 'Severity Algorithm Validation' subsection for the first time. Given to limited resources to replicate the severity model in a separate primary care database, this data split is an important step and aims to conduct the survival analysis in an independent sample of the data to validate the developed severity model versus adverse outcomes and mortality.
- To summarise, the 'Diabetes Severity Algorithm' subsection now describes the random sample split and that the severity algorithm to be developed in 90% of the study cohort, two general approaches to be used to derive severity scores, the regression model to be used to calculate the domains' weights, and finally our plan to account for domains with multiple measurements and missing data. To continue the chronological order of our analyses steps, the following 'Severity Algorithm Validation' subsection clearly states the use of a statistical approach for validating the developed algorithm, lists the pre-defined adverse outcomes to be used in this validation analysis, re-states that the validation exercise to be conducted in 10% of the diabetes cohort, then mentions that survival models will be based on calculated subdomains weights that were derived by the regression model estimates.

Changes:

- i. The Abstract now includes the following edits to indicate the model used to develop the severity score and the model used to validate the score:

"Regression models estimates will be used to calculate domains weights. Survival analysis for the association between weighted severity scores and future outcomes....."

ii. The plan to randomly split the study cohort into a training and validation datasets is introduced earlier in the protocol, importantly in the "Study Population" subsection as follows:

"A random 90% of the identified diabetes cohort (training dataset) will be used to develop the severity tool, with the remaining 10% of the cohort used as a validation dataset, as described below."

iii. The 'Diabetes Severity Algorithm' subsection is edited as follows:

- "Using annual data bins and grouping diabetes patients in the training dataset (include random 90% of the total diabetes cohort) from 1st April to 31st March between 2007 to 2017, the developed diabetes algorithm will grade the severity of T2DM using pre-defined (sub)domains."
- "Then, regression model will be fitted from which the weights of its estimates will be used to calculate the weights for severity sub-domains."

iv. The 'Severity Algorithm Validation' subsection is edited as follows:

"As stated earlier, a random 10% of the study cohort (validation dataset) will be used to validate the performance of the severity algorithm that was developed in the training dataset (90% of the diabetes cohort). Using the calculated subdomains weights (derived by the regression model estimates), Cox regression analysis will be used to validate the developed severity algorithm and we will assess the model performance"

VERSION 3 – REVIEW

REVIEWER	Rosa Gini Agenzia regionale di sanità della Toscana, Italy
REVIEW RETURNED	15-Mar-2018

GENERAL COMMENTS	I thank the authors for effectively addressing my previous comments. There is a last important point that is not clear to me in the 'Diabetes Severity Algorithm' subsection of the Methods section: what is the dependent variable that will enter the regression that will be ran to estimate the weights for severity sub-domains? My understanding from the current text is that the dependent variable will be a score of severity computed as a weighted combination of the severity sub-domains themselves, using clinical input to score them with 1, 2 or 3. But of course this would be circular, and the regression would then simply re-estimate the input scores. So what is it truly meant here? Probably a practical example would help clarifying? Beyond this small point, I have no further comments, and I send to the authors my best wishes for their study!
--

VERSION 3 – AUTHOR RESPONSE

1. I thank the authors for effectively addressing my previous comments.

There is a last important point that is not clear to me in the 'Diabetes Severity Algorithm' subsection of the Methods section: what is the dependent variable that will enter the regression that will be ran to estimate the weights for severity sub-domains? My understanding from the current text is that the dependent variable will be a score of severity computed as a weighted combination of the severity sub-domains themselves, using clinical input to score them with 1, 2 or 3. But of course this would be circular, and the regression would then simply re-estimate the input scores. So what is it truly meant here? Probably a practical example would help clarifying?

Beyond this small point, I have no further comments, and I send to the authors my best wishes for their study!

Authors' response:

Thank you for the time reviewing our paper again and for the kind wishes. We agree with the reviewer that this point needs to be further clarified in the paper.

The dependent variable in the regression that will be ran to estimate the weights for severity sub-domains will be death (primary outcome) and hospitalisation (secondary outcome). Apologies if this was not satisfactorily clear in the text, but the score of severity computed as a weighted combination of the severity sub-domains will be severity score covariate (independent variable) and not the dependent variable in the regression model.

Change:

The following text has been added to the 'Diabetes Severity Algorithm' subsection in response to this comment:

"Then, regression model, using death (primary outcome) and future hospitalisation (secondary outcome) as dependent variable, will be fitted from which the weights of its estimates will be used to calculate the weights for severity sub-domains."

2. Editorial Requirements

Please remove the conclusions section, as this is not required in protocol articles. Instead, please include an ethics and dissemination section.

Change:

We thank the editorial team for their comment. The 'Conclusions' section has been replaced with an 'Ethics and Dissemination' section.

VERSION 4 – REVIEW

REVIEWER	Rosa Gini Agenzia regionale di sanità della Toscana
REVIEW RETURNED	26-Apr-2018
GENERAL COMMENTS	type on page 10: 'Then, regression model,' -> 'Then, regression models,' Good luck with your study!